

# Measurement report: CCN activity and its variation with organic oxidation level and volatility observed during aerosol life cycle intensive operational period (ALC-IOP)

Fan Mei[1], Jian Wang[2], Shan Zhou[3,4], Qi Zhang[3], Sonya Collier[3,5] and Jianzhong Xu[3,6]

[1]Pacific Northwest National Laboratory, Richland, WA, 99352, USA
[2]Washington University in St. Louis, St. Louis, MO, 63130, USA
[3]Department of Environmental Toxicology, University of California, 1 Shields Ave., Davis, CA, 95616, USA
[4]Currently at Department of Civil and Environmental Engineering, Rice University, Houston, TX, 77005, USA

[5]Currently at California Air Resources Board, 1001 I Street, Sacramento, CA, USA
[6]State Key Laboratory of Cryospheric Science, Northwest Institute of Eco-Environment and Resources, Chinese Academy of Science, Lanzhou, Gansu, 730000, China

*Correspondence to*: Fan Mei (fan.mei@pnnl.gov), Jian Wang (jian@wustl.edu)

**Abstract.** Cloud condensation nuclei (CCN) spectrum and the CCN activated fraction of size selected aerosols (SR-CCN) were measured at a rural site on Long Island during the Department of Energy (DOE) Aerosol Life Cycle Intensive Operational Period (ALC-IOP) from July 15 to August 15, 2011. During the last week of the ALC-IOP, the dependence of the activated fraction on aerosol volatility was characterized by sampling downstream of a thermodenuder operated at temperatures up to 100 ℃. Here we present aerosol properties, including aerosol total number concentration, CCN spectrum, and the CCN hygroscopicity for air masses of representative origins during the ALC-IOP. The hygroscopicity of organic species in the aerosol is derived from CCN hygroscopicity and chemical composition. The dependence of organic hygroscopicity on the organic oxidation level (e.g., atomic O:C ratio) agrees well with theoretical predictions and results from previous laboratory and field studies. The derived $\kappa_{org}$ and O:C ratio first increase as thermal denuder (TD) temperature increases from 20 ℃ (i.e., ambient temperature) to 50 or 75 ℃, then decreases as TD temperature further increases to 100 ℃. These trends are different from previous laboratory experiments and field observations, which reported that organic O:C increased monotonically with increasing TD temperature, whereas $\kappa_{org}$ decreased with the TD temperature. The initial increases of O:C and $\kappa_{org}$ with TD temperature below 50 ℃ are likely due to the evaporation of more volatile organics with relatively lower O:C and hygroscopicity such as primary OA. Previous studies were either focused on laboratory-generated SOA or based on field observations at locations dominated by SOA.



## 1. Introduction

As a critical element in cloud formation, atmospheric aerosols indirectly influence the global energy budget by affecting the atmospheric boundary structure and changing clouds' lifetime and coverage. An increase in cloud condensation nuclei (CCN) concentration leads to smaller cloud droplet sizes and higher cloud albedo at the same liquid water path (i.e., first indirect effect, Twomey, 1977). Additionally, smaller cloud droplets suppress precipitation, causing increases in cloud lifetime and coverage (i.e., second indirect effect, Albrecht, 1989). Presently, the aerosol indirect effects remain the most uncertain components in simulated radiative forcing since the pre-industrial era (Pachauri et al., 2014). This considerable uncertainty is due to an incomplete understanding of the aerosol-cloud interactions and the perturbation of aerosol properties due to anthropogenic emissions (Rosenfeld et al., 2014).

Quantifying the aerosol indirect effects requires the knowledge of aerosol particles' ability to form cloud droplets at atmospherically relevant supersaturations (i.e., cloud condensation nuclei (CCN) activity). Under a given supersaturation, whether a particle can activate and form a cloud droplet depends on its size and hygroscopicity parameter (Andreae and Rosenfeld, 2008a;Petters and Kreidenweis, 2007;Wang et al., 2008). The Hygroscopicity parameter describes the tendency of particles to uptake water. It is a function of thermodynamic properties, including molar volume, activity coefficient, and surface activity, of the particles' species (McFiggans et al., 2006). The number of inorganic species in ambient aerosol particles is quite limited, and their hygroscopicities are well understood (Petters and Kreidenweis, 2007). On the other hand, atmospheric aerosol often consists of a large number of organic species (Zhang et al., 2007). Collectively, these organic species frequently dominate the composition of sub-micron particles, which make up the majority of CCN. The hygroscopicity parameters of these organic species may depend on water solubility, molecular weight, oxidation level, surface activity, and/or phase state (Kuwata et al., 2013;Lambe et al., 2011;Mei et al., 2013a;Ovadnevaite et al., 2017;Wang et al., 2019), and exhibit a wide range of values from zero for hydrophobic chemical species to ~0.3 for some of low molecular weight water soluble organics (e.g., Petters et al., 2009;Moore et al., 2012;Lathem et al., 2013).

Earlier studies show that the simulated CCN concentration can be strongly correlated to the hygroscopicity of organics in the particles (e.g., McFiggans et al., 2006;Mei et al., 2013b) in addition to particle size distribution (Dusek et al., 2006), mixing state (Lance et al., 2013;Mei et al., 2013c;Wang et al., 2010), and volume fraction of organics in particles (Wang et al. 2008). The high sensitivity of simulated CCN concentration to organic hygroscopicity is more prevalent during the pre-industrial era when anthropogenic sulfate concentration was lower, and organics represented a substantially larger fraction of the submicron aerosol mass (Mei et al., 2013b;Liu and Wang, 2010). As a result, neglecting the variation in organic hygroscopicity can lead to substantial bias in aerosol indirect forcing estimation (i.e., the change in radiation flux due to the increased aerosol concentration since the pre-industrial era) (Liu and Wang, 2010). These results highlight the importance of understanding organic hygroscopicity variability and accurately representing it in climate models. Earlier studies show that organic hygroscopicity for CCN activation generally increases with oxidation level (Duplissy et al., 2011;Lambe et al., 2011;Massoli et al., 2010;Mei et al., 2013a;Mei et al., 2013c;Thalman et al., 2017), suggesting a promising approach to efficiently





parameterize the overall hygroscopicity of a large number of organic species in aerosol particles. Wang et al. (2019) show that

for secondary organic aerosols (SOA), the observed increasing organic hygroscopicity with oxidation level is likely due to the following two reasons. First, SOA formed from smaller precursor molecules are more oxidized and have lower average molecular weight. Second, fragmentation reactions during oxidation reduce average organic molecule weight, leading to increased hygroscopicity. At present, field measurements of organic hygroscopicity and its variation with oxidation level are still quite limited. The formation of SOA strongly depends on the volatility of organic species. There have been very few

studies on the variation of hygroscopicity with both oxidation level and volatility of organics, especially for ambient aerosols (Cerully et al., 2015;Kuwata et al., 2011).

Here, we report aerosol hygroscopicity, mixing state, and organic hygroscopicity derived from size-resolved CCN activated fraction and chemical composition measured at a rural site on Long Island during the Aerosol Life Cycle Intensive Operational Period (ALC-IOP). Aerosol properties are presented in Section 3.1 for representative air mass types, which are classified based

on the analysis of air mass back-trajectories (Zhou et al., 2016). The hygroscopicity of activated aerosol particles is presented for each airmass type, and the influence from the characteristic emission sources and atmospheric processing is discussed. In section 3.2, the hygroscopicity of organic species in the aerosol is derived from particle hygroscopicity and chemical composition. The variation of organic hygroscopicity with air mass type is presented. The relationship between organic hygroscopicity and oxidation level is examined, and the relationship is compared with results from earlier studies. Finally, in

Section 3.3, the variation of organic hygroscopicity with volatility and oxidation level is studied using the measurements downstream of a thermal denuder operated at temperatures ranging from ambient temperature to 100ºC.

## 2. Experimental method

### 2.1. Measurements and the site

The Aerosol Life Cycle Intensive Operational Period (ALC-IOP), a field study sponsored by the Department of Energy (DOE),

took place at Brookhaven National Laboratory (BNL, 40.871˚N, 72.89˚W) on Long Island, New York from July 15 to August 15, 2011. Aerosol properties at this location were influenced by complex interaction among anthropogenic, biogenic, and marine emissions with the extent of atmospheric processing, which also depended on air mass trajectories and atmospheric transport time. The measurements related to this study include CCN activated fraction of size-classified particles, CCN spectrum, aerosol size distribution, and size-resolved chemical composition of non-refractory submicron aerosols (NR-PM$_1$).

Measurements included both ambient aerosols and those processed by a digitally-controlled thermodenuder (TD) to examine the variation of aerosol properties with volatility (Zhou et al., 2016). All of the above measurements were taken at the ground level and are reported at ambient conditions, and local time (UTC minus 4 hrs) is used throughout this study.





### 2.2. Size-resolved CCN activated fraction, CCN spectrum, and aerosol size distribution

The CCN activated fraction was measured using a size-resolved CCN counter system (SR-CCN). The SR-CCN is detailed in
Mei et al. (2013a, 2013b) and is briefly described here. Ambient aerosol particles are first dried to below 20% relative humidity
(RH), brought to a steady-state charge distribution in a Kr-85 aerosol neutralizer (model 3077, TSI), and subsequently
classified by a differential mobility analyzer (DMA, model 3081, TSI). The total number and CCN concentrations of the size-
classified aerosol are then simultaneously characterized by a condensation particle counter (CPC, model 3071, TSI) and a CCN
counter (CCN-100, DMT), respectively (Roberts and Nenes, 2005;Lance et al., 2006;Rose et al., 2008). During the ALC-IOP,
the DMA's aerosol sample flow and sheath flow were maintained at 0.8 and 8 L min$^{-1}$, respectively. The total flow of CPC was
reduced to 0.5 L min$^{-1}$ within a critical orifice inline and the sample flow of the CCN counter was maintained at 0.3 L min$^{-1}$.
The measurement sequence is illustrated in Figure S1 (Supplementary Information). The longitudinal temperature gradient of
the CCN counter was stepped through 4, 4.5, 5.5, 6.5, 8, 10, 12, 18°C (Fig. S1b). Based on calibrations using ammonium
sulfate particles, the corresponding supersaturations ($S$) derived from $\kappa$–Köhler theory (using a constant van't Hoff factor of
2.5) were 0.12%, 0.15%, 0.20%, 0.25%, 0.32%, 0.41%, 0.50% and 0.79%, respectively. The supersaturation was maintained
inside the CCN counter at each value for approximately 9 minutes. The classified particle diameter (by the DMA) was scanned
between 60 nm and 250 nm four times (128 seconds per scan) (Wang and Flagan, 1990).  The CCN counter was stepped
through the temperature gradients in a "sawtooth" pattern (Fig. S1b) and provided measurements at the above eight $S$ values
approximately every 80 minutes. Before July 23, the CCN counter was operated with the first seven supersaturation setpoints.
The aerosol size distribution was derived by inverting the particle concentration measured by the CPC using a routine described
in Collins et al. (2002). A similar procedure was also applied to measured CCN concentration to obtain size-resolved CCN
concentration. The ratio of the above two concentrations provided size-resolved CCN activated fractions ($E$). In addition to
size-resolved CCN activated fraction, CCN concentration spectrum was measured using a second CCN counter also operated
at a flow rate of 0.3 L min$^{-1}$, and at seven supersaturations of 0.11%, 0.13%, 0.17%, 0.23%, 0.32%, 0.40% and 0.48%
corresponding to longitudinal temperature gradients of 4.3, 4.8, 5.5, 6.5, 7.9, 10, and 12 °C, respectively. The temperature
gradient was stepped through the eight values every 32 minutes, as shown in Figure S1a.

The measurement of the CCN concentration spectrum was carried out for ambient aerosol during the entire ALC-IOP. Before
August 10, the SR-CCN sampled ambient aerosol only. From August 10 to 15, 2011, the SR-CCN was operated downstream
of the TD (custom). The design and operation of the TD were improved by Fierz (Fierz et al., 2007) and were also discussed
by Zhou et al. (2016). During this period, the DMA inside SR-CCN was scanned between 80 nm and 250 nm four times (66
secs per scan) at each supersaturation setpoint. Every 40 minutes, the measurements were stepped through 7 supersaturations,
similar to the measurement sequence before August 10, except without the setpoint at 0.41% (i.e., the temperature gradient of
10°C). From August 10 to 15, The temperature setting of the TD was cycled through 50, 75, and 100 ℃. The TD was operated
with an automated switching valve to allow for measurements of ambient aerosol particles (bypass (BP) mode) for 20 mins



and then processed by the TD (TD mode) for 20 mins. A complete cycle included ambient aerosol measurements, TD processed

measurements at 50 ℃ ( or 75 ℃, or 100 ℃) alternatively.

The number size distribution of ambient aerosol was measured with a custom-made scanning mobility particle spectrometer (SMPS) during the entire ALC-IOP. The SMPS consists of an $^{85}$Kr neutralizer (Model 3077A, TSI Inc), a DMA (Model 3080, TSI Inc.), and a CPC (Model 3771, TSI, Inc.). The SMPS was operated for the mobility diameter range of 10 - 610 nm with

a time resolution of 120 seconds. The aerosol size distribution was inverted using the routine described in Collins et al. [2002], which explicitly accounts for multiply charged particles.

.

### 2.3.  Volatility-resolved chemical composition of submicron aerosols

The mass concentrations of NR-PM$_1$ organic and inorganic (nitrate, sulfate, ammonium, chloride) species and their volatility

distributions were measured by combining the HR-ToF-AMS (Aerodyne Research Inc., from now on AMS for short) (DeCarlo et al., 2006) with the TD. The AMS sampled ambient aerosols and those processed by the TD alternately during the entire ALC-IOP. Positive Matrix Factorization (PMF) analysis of the AMS organic matrix resolved three distinct organic aerosol (OA) factors, including (1) a fresher, semivolatile oxygenated organic aerosol (SV-OOA; O/C = 0.54; 63% of OA mass) representing SOA formed through the interactions between local biogenic VOCs and anthropogenic emissions in transported

urban plumes from the New York metropolitan region; (2) a regional, more aged, low-volatility OOA (LV-OOA; O/C = 0.97; 29%) influenced by aqueous-phase processing; and (3) a nitrogen-enriched OA (NOA; O/C = 0.19; N/C = 0.185; 8%) likely composed of amine salts formed from acid-base reactions in industrial emissions (Zhou et al., 2016). The volatilities of the OA were determined based on their thermal profiles. Details of the AMS operation, data reduction and processing, PMF analysis, and data interpretation are available in Zhou et al. (2016).

## 3.   Discussions and results

### 3.1.  Overview of aerosol properties in different air masses

The back-trajectories of air masses arriving at the site were classified into five clusters (Zhou et al., 2016). Because the aerosols originated from the North Atlantic Ocean area likely had a substantial contribution from sea spray particles that could not be quantitively measured by the AMS, the analysis of CCN activity and its relationship with the aerosol composition are focused

on the other four clusters, which are briefly described here. The air mass of cluster 1 had a relatively long-range transportation influence from the northwest Canadian forest (LRNW, 15.1% of all trajectories). The air mass of cluster 2 was mainly from the northwest (NW, 21.6%) region. Cluster 3 included air mass circulating along the south-southwest, with some passing over Philadelphia and Washington metropolitan areas (SSW, 26.4%). Cluster 4 represents the air mass passing over the polluted west NYC metro area (W, 14.8%).





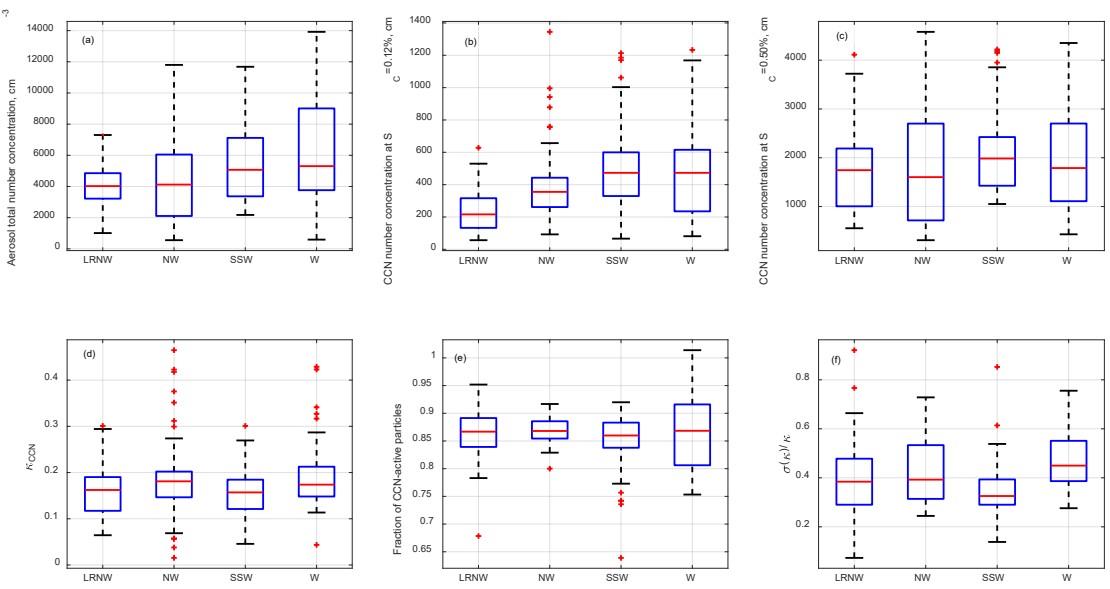


**Figure 1. The averaged aerosol properties were observed for each air mass cluster at the BNL site from July 15 to August 9, 2012. (a) aerosol total number concentration; (b) CCN concentrations measured at supersaturations of 0.12% and (c) 0.48%; the derived properties for the CCN active particles ranging from 80 to 250 nm (d) overall aerosol hygroscopicity; (e) maximum activation fraction, and (f) dispersion of size-selected CCN active particles. The ends of the whiskers represent the minimum and maximum**
**of data except for the outlines (red cross), which are defined as points outside of ±2.7σ. The bottom and the top of the box are the 25th and 75th percentiles, the line inside the box is the 50th percentile.**

Fig. 1a shows the total aerosol number concentration, integrated from the aerosol size distribution ranging from 10 to 610 nm measured by the SMPS, and Fig. 1b and 1c present the measured CCN concentrations at two supersaturations of 0.12% and 0.50%, respectively. The CCN concentration at a supersaturation of 0.12% is dominated by accumulation mode particles,
whereas both Aiken mode and accumulation mode particles contribute to the CCN at 0.50% supersaturation. The highest total number concentration was observed when the airmass was from the West (i.e., W cluster) following by the airmasses from SSW, mainly due to the strong anthropogenic emissions in the New York metropolitan and Boston metropolitan areas. Figure 1b indicates that the airmasses in W and SSW clusters also had the highest average accumulation mode number concentration. The trends from both plots are consistent with the trend of aerosol mass loading measured by the AMS (Zhou et al., 2016).

In contrast, the 25%- 75% percentiles of CCN concentrations at 0.5% supersaturation ranged from 1,000 to 2,500 cm$^{-3}$ for all clusters, indicating no clear trend with air mass observed for CCN concentration at 0.12%. These results suggest a strong size-dependency in the CCN activation properties in the aerosol particles from different regions. Only for the larger accumulation mode aerosol particles, the air masses from the Pennsylvania metropolitan and New York metropolitan areas contain more CCN active particles than aerosol particles from other origins.



Particle hygroscopicity is derived from the size-resolved CCN activated fraction following the approach detailed in Mei et al. (2013a and 2013b). In brief, the characteristic critical supersaturation ($S^*$) of the size selected CCN is defined as the supersaturation at which the activated fraction reaches 50% of the maximum activation fraction ($E$). The valve of 1-$E$ represents the number fraction of non-CCN (e.g., particles consisting of non-hygroscopic species only) for the size-selected particles. The median hygroscopicity of the CCN ($\kappa_{CCN}$) is given by:

$$\kappa_{CCN} = \frac{4A^3}{27D_p^3(S*)^2} \tag{1}$$

Where $A = \frac{4\sigma_w M_w}{RT\rho_w}$, $M_w$ represents the molecular weight of water, $\sigma_w$ is the surface tension of pure water, $\rho_w$ is the density of water, $R$ the gas constant, and $T$ the absolute temperature. The dispersion of CCN hygroscopicity (Fig. 1f) is defined as $\sigma(\kappa)/\bar{\kappa}$, where $\sigma(\kappa)$ and $\bar{\kappa}$ are the standard deviation and the average of the CCN hygroscopicity, respectively. The value of $\sigma(\kappa)/\bar{\kappa}$ reflects the heterogeneity in hygroscopicity and composition of the activated particles. A lower $\sigma(\kappa)/\bar{\kappa}$ Suggests more

homogeneous particle composition as in internally mixed aerosols.

The statistics of $\kappa_{CCN}$ for particles ranging from 88 to 192 nm are shown for each of the four airmass clusters in Fig. 1d. While $\kappa_{CCN}$ shows a wide range of values from near zero to 0.5, the 25%-75% percentiles of the $\kappa_{CCN}$ values do not exhibit any significant differences among the four clusters and are between ~0.1 and ~0.2. The median value of $\kappa_{CCN}$ is ~ 0.15 for all four clusters, substantially below 0.3 suggested for continental aerosol (Andreae and Rosenfeld, 2008b). The median $E$ value is

~87% for the four clusters, suggesting aerosols observed had a relatively minor contribution from freshly emitted non-hygroscopic particles. The W and NW clusters have the largest and the smallest variabilities in $E$ values, respectively. The $E$ values indicate that aerosols in the long-range transported NW air masses (i.e., LRNW cluster) were not all internal mixtures and included some contribution of freshly emitted non-hygroscopic aerosol particles. Statistically, the SSW aerosols show the lowest hygroscopicity dispersion (Fig. 1f), suggesting that CCN had similar chemical composition and were likely more aged

at given sizes. The variety of aerosol sources along the LRNW trajectory paths likely contribute to the relatively large variability of the aerosol hygroscopicity dispersion for the cluster.

### 3.2. Relationship between $\kappa_{org}$ and organic oxidation level

The relationship between the hygroscopicity of organic species in the aerosol particles ($\kappa_{org}$) and the average oxidation level (i.e., atomic O:C ratio) is examined using the ALC-IOP measurements. On average, aerosols observed during the IOP were

neutralized, and BC represented a negligible fraction of the total submicron aerosol volume (1.9%). It is expected that sea salt had a minor contribution to the submicron aerosol composition for airmasses of the four clusters. Here we assume submicron aerosols consisted of ammonium sulfate, ammonium nitrate, and organics. The organic hygroscopicity $\kappa_{org}$ is therefore derived by subtracting the contribution of the sulfate and nitrate from the CCN hygroscopicity $\kappa_{CCN}$:



$$\kappa_{org} = \frac{1}{x_{org}} \left( \kappa_{CCN} - x_{(NH_4)_2SO_4} \kappa_{(NH_4)_2SO_4} - x_{NH_4NO_3} \kappa_{NH_4NO_3} \right) \tag{2}$$

where $x_i$ is the volume fraction of species $i$ at the exact particle size of $\kappa_{CCN}$. Furthermore, $\kappa$ values are 0.67 and 0.61 for

(NH$_4$)$_2$SO$_4$ and NH$_4$NO$_3$, respectively. To increase the signal-to-noise ratio and reduce the uncertainty in derived $\kappa_{org}$ (Mei et

al., 2013b), we average $x_{(NH_4)_2SO_4}$ and $x_{NH_4NO_3}$ over periods during which particle composition showed minimal variations

and was dominated by organics. The criteria used to identify these periods include average $x_{org}$ above 60% and are detailed in

Mei et al., (2013b). A total of 47 such periods is identified.  During 35 of the 47 periods, air mass showed consistent cluster

type, and the periods are denoted as one of four clusters. The size-resolved $\kappa_{org}$ at particle diameter ranging from 103 to 181

nm is first derived using Eq. (2) for the 35 periods and then averaged for each of the four cluster types. Size-resolved organic

O:C ratios at the same diameter range are calculated from AMS measurements.The aerodynamic aerosol size measured by

AMS was converted to the aerosol mobility size (Mei et al., 2013a;Mei et al., 2013c). The variation of the average $\kappa_{org}$ with

the corresponding O:C ratio is shown in Figure 2. The uncertainty of derived $\kappa_{org}$ is derived following the same approach

detailed in Mei et al. (2013b), which is based on error propagation and the uncertainties of the variables in Eq. (2). The

uncertainty in derived O:C was estimated as 10% (Zhou et al., 2016). Figure 2 shows that the average $\kappa_{org}$ generally increases

with the O:C ratio, and the variation follows the trend reported by an earlier laboratory study (Lambe et al. 2011), but with a

slightly different offset in the derived $\kappa_{org}$.  Note that the O:C ratios from Lambe et al. (2011) were scaled by a factor of 1.27

to account for changes in the method of calculating the O:C ratio (Canagaratna et al., 2015). During the ALC-IOP, organic

aerosol observed mainly was secondary. Wang et al. (2019) show that for SOA, organic hygroscopicity is not limited by water

solubility and is mainly controlled by the molecular weight of the organic species. The increase of SOA hygroscopicity with

O:C is primarily due to the combination of two effects. One effect is that the SOAs formed from smaller precursor molecules

tend to have lower average molecular weight and be more oxidized. The other happens during oxidation; fragmentation

reactions reduce average organic molecule weight, leading to increased hygroscopicity. The variation of $\kappa_{org}$ with O:C can be

predicted provided the volatility of SOA is known. The variation of cluster average $\kappa_{org}$ with O:C agrees with those predicted

by Wang et al., (2019) for $\log_{10}C^*$ values between -5 to -1, a typical range for ambient SOA. Here $C^*$ (μg/m$^3$) is the organic

volatility.





**Figure 2. Derived $\kappa_{org}$ as a function of O:C atomic ratio for four clusters, and the $\kappa_{org}$ values of SOA positive matrix factorization (PMF) factors (LV-OOA, SV-OOA, NOA) derived from measurements during the ALC-IOP. Also shown with the predicted variation for organics with a mean logC* equals to -1 and -5, and the $\kappa_{org}$ values of SOA PMF factors from measurements during GoAmazon 2014/5 (Thalman et al., 2017), and the relationship between $\kappa_{org}$ and O:C for SOA derived from a laboratory study (Lambe et al., 2011).**

The organic species were classified into three SOA factors based on the PMF analysis, including a fresh semivolatile oxygenated organic aerosol (SV-OOA), which contributed 63% of OA mass and was strongly influenced by urban plumes transported from the W and SSW regions, a regional and more aged low-volatility oxygenated organic aerosol (LV-OOA), which was influenced by aqueous-phase processing, and a nitrogen-enriched OA (NOA), which likely composed of amine





salts formed from acid-base reactions in industrial emissions (Zhou et al., 2016). The mass fractions of the three factors for each of 47 periods are shown in Fig. 3. The first 25 periods are before August 2, 2011, and NOA substantially contributed to

the aerosol mass concentration. The fraction of NOA during the remaining periods (after August 2, 2011, from periods 26-47) is less than 2% and negligible. For the same air mass cluster type, LV-OOA fractions are generally higher during the periods after August 2, 2011, when the site experienced more cloudy conditions and precipitation. The elevated LV-OOA fraction is likely due to a more substantial influence by aqueous-phase processing (Zhou et al., 2016).

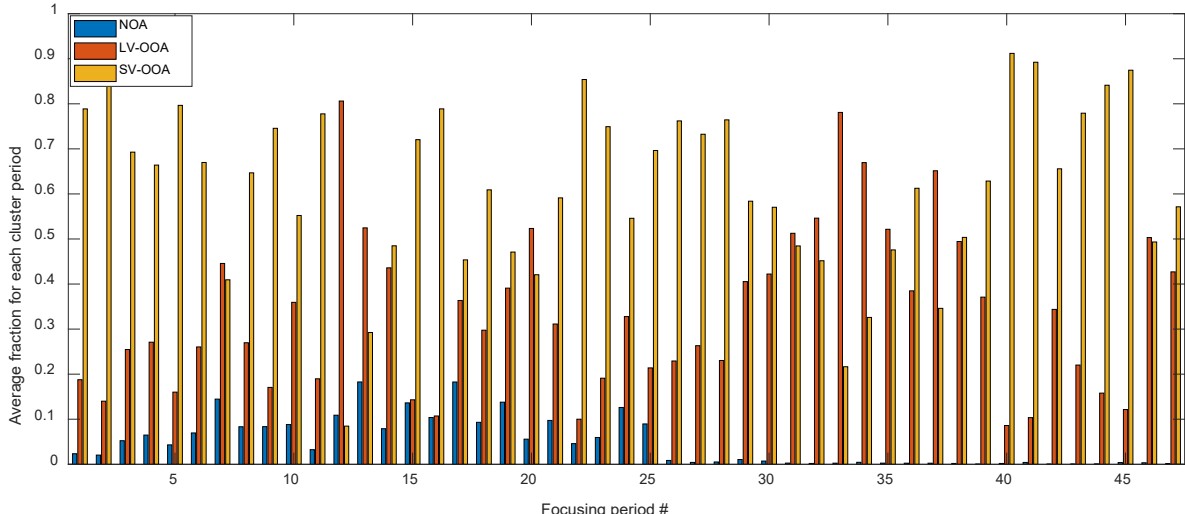

**Figure 3. Average mass fraction of each PMF factor for the 47 focus periods.**

The three PMF factors are associated with different emission sources and atmospheric processes (Zhang et al., 2011;Zhou et al., 2016). Hygroscopicities of the three organic factors ($\kappa_{LV\text{-}OOA}$, $\kappa_{SV\text{-}OOA}$, and $\kappa_{NOA}$) were retrieved using multilinear regression of $\kappa_{org}$ to the volume fractions of the organic factors (Thalman et al., 2017). This regression is based on the Levenberg-Marquardt algorithm and yields the hygroscopicities for LV-OOA, SV-OOA, and NOA as 0.170, 0.069, and 0.072,

respectively. The O:C ratios of LV-OOA, SV-OOA, and NOA calculated using the Improved Ambient (IA) method (Canagaratna et al., 2015) are 0.97, 0.54, and 0.19, respectively. The retrieved $\kappa_{LV\text{-}OOA}$, $\kappa_{SV\text{-}OOA}$, and $\kappa_{NOA}$, show a general increasing trend with increasing O:C ratio (Fig. 2, purple squares), in agreement with those derived from an earlier field study (Thalman et al., 2017) and predicted by Wang et al. (2019). We note $\kappa_{SV\text{-}OOA}$ is lower than the model prediction and the SOA hygroscopicity at the same O:C ratio reported by Lambe et al. 2011. One possible explanation is that the SV-OOA was mainly

composed of organic compounds formed through oxidation of urban emissions, thus was enriched with oxidized yet relatively hydrophobic hydrocarbons (Zhou et al., 2016). As HOA are typically hydrophobic, the inclusion of HOA likely leads to a $\kappa_{SV\text{-}OOA}$ value below the predicted and reported SOA hygroscopicity (Lambe et al., 2011, Wang et al., 2019).





### 3.3. Volatility-resolved hygroscopicity of activated aerosol particles

The bulk volumetric fractions of major species, including sulfate, nitrate, and organics, and the O:C ratio are shown in Fig. 4
as a function of time from August 10 to August 15 for ambient aerosol and those processed by the TD at 50 °C, 75 °C, and 100 °C. During the week, nitrate had a negligible contribution to aerosol composition (i.e., less than 5% in volume). Because sulfate is non-volatile at temperatures below 100 °C and some semivolatile organics were evaporated inside the TD, ammonium sulfate fraction increases with TD temperature.   The variation of organic hygroscopicity with volatility is examined using measurements during four periods. Period 1 is from 08/11/08:00 to 08/11/19:10; period 2 is from 08/11/2011 19:10:00 to
08/12/2011 08:40:00; period 3 is from 08/12/2011 08:40 to 08/13/2011 17:10:00; and period 4 is from 08/14/2011 06:40:00 to 08/14/2011 22:32:00.   These periods are chosen because of the high organic volume fraction (i.e., greater than 65%, Mei et al. 2013a and 2013b) and relatively constant particle composition and volatility. Aerosol properties, including particle hygroscopicity and species volume fractions, are averaged for the four periods. The derivation of the particle hygroscopicity parameter $\kappa_{CCN}$ and $\kappa_{org}$ for each period follows the same approach described in sections 3.1 and 3.2, except that species volume
fractions (i.e., $x_{(NH_4)_2SO_4}$ and $x_{NH_4NO_3}$) are based on the bulk measurements due to the operation mode change in the TD period.

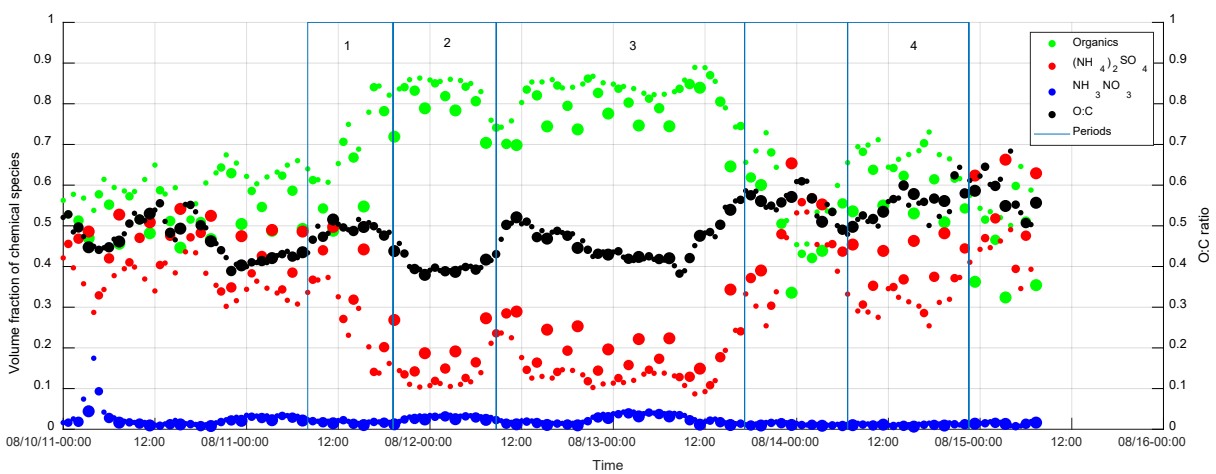

**Figure 4.  Time series of the volume fractions of organics, (NH₄)₂SO₄, and NH₄NO₃ observed at the BNL site from August 10 to August 15, 2012.; Note that the increase of the symbol size represents the TD temperature increase from the ambient temperature**
**to 100 °C. The green dot represents the organic species. The red dot represents ammonium sulfate. The blue dot represents ammonium nitrate. The black circle represents the O:C ratio.**

As shown in Fig. 5a, $\kappa_{CCN}$ increases with increasing TD temperature, mainly due to the increased volume fraction of the ammonium sulfate. The TD processed aerosol particles with a higher inorganic fraction (period 1 and 4) exhibited a more significant change of CCN hygroscopicity ($\Delta\kappa_{CCN}$=~0.10) compared to the other two periods ($\Delta\kappa_{CCN}$=~0.05) with the increase





of TD temperature (Fig. 5c). The derived $\kappa_{org}$ first increases as TD temperature increases from 20 °C (i.e., ambient temperature) to 50 or 75 °C, then decreases as TD temperature further increases to 100 °C. The O:C ratio shows a similar trend except during Period 1, when O:C decreased monotonically with TD temperature due to the contribution of NOA. The variations of $\kappa_{org}$ and O:C with TD temperature setting observed during ALC-IOP are different from previous laboratory experiments and field observations, which reported that organic O:C generally increased with increasing TD temperature (Huffman et al.,

2009;Kuwata et al., 2011)  whereas $\kappa_{org}$ decreased monotonically with TD temperature (Asa-Awuku et al., 2009;Cain and Pandis, 2017;King et al., 2009;Kuwata et al., 2011). The heating inside the TD can lead to both evaporation and reaction (e.g., oligomerization) of particle-phase organic species. Whereas oligomerization is not expected to influence O:C appreciably, it leads to lower $\kappa_{org}$ values due to the increase of molecular weight. Therefore, oligomerization likely played a relatively minor role when processed by the TD temperature at 50 °C. The initial increases of O:C and $\kappa_{org}$ at TD temperature below 50 °C is

likely due to the evaporation of more volatile organics with relatively lower O:C and hygroscopicity, such as primary OA. The evaporation of primary OA can also explain the difference in the dependences of $\kappa_{org}$ with TD temperature as previous studies were either focused on laboratory-generated SOA(Asa-Awuku et al., 2009;Cain and Pandis, 2017;King et al., 2009;Kuwata et al., 2011) or based on field observations at locations dominated by SOA (Cerully et al., 2015;Saha et al., 2017). The initial increase of O:C and $\kappa_{org}$ could also be partially due to the evaporation of SOA with lower O:C. As $\kappa_{org}$ of SOA mainly depends

on molecular weight, the simultaneous increases of O:C and $\kappa_{org}$ would indicate evaporation of more volatile secondary organics with relatively large molecule weight and low O:C. At the upper TD temperature range (i.e., above 50 and 75 °C), the decrease of O:C indicates the evaporation of more oxygenated organics. As the addition of oxygenated function groups reduces the volatility of organic species, the organics with relatively high O:C evaporated at a high TD temperature setting is expected to have a smaller molecular weight (i.e., molecule size) compared to those remaining in the particle phase, consistent

with the decreasing $\kappa_{org}$. While we have no direct evidence, oligomerization at high temperature may also contribute to the decreasing of $\kappa_{org}$. The variations of $\kappa_{org}$ and O:C with TD temperature setting depends on the distributions of oxidation level, molecular size, and volatility of organic species in the particle phase, which can lead to contrasting trends at different temperature ranges. These distributions can vary substantially with aerosol type and sampling location, leading to different trends in the variation of $\kappa_{org}$ and O:C with the TD temperature setting.





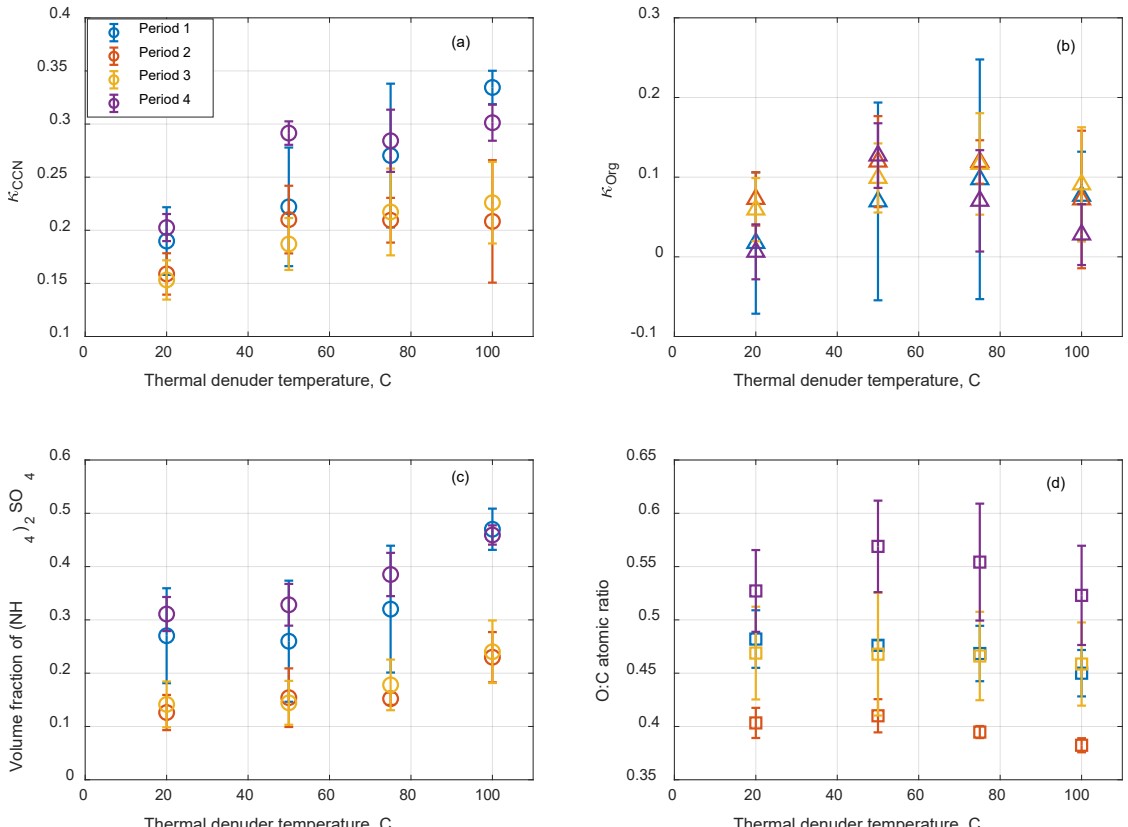

**Figure 5. Averaged aerosol hygroscopicity($\kappa_{CCN}$) (a), the derived $\kappa_{org}$ (b), the volume fraction of (NH₄)₂SO₄ (c), and O:C ratio (d) as a function of the thermal denuder temperature. The error bar represents the standard deviation of the measurand values during each period.**

## 4.  Conclusions

This work focuses on the CCN activity and its variations with organic oxidation level (O:C ratio) and volatility using measurements at the Brookhaven National Laboratory from July 15 to August 15, 2011, during ALC-IOP. Aerosol properties, including aerosol total number concentration, CCN spectrum, and the CCN hygroscopicity, are examined for four air mass clusters, representing different ambient aerosol emission sources, transformation pathways, and atmospheric processes. Aerosols originated from the Pennsylvania metropolitan and New York metropolitan areas contained more CCN active particles than aerosols in LRNW and NW airmass clusters. The 25%-75% percentiles of the CCN hygroscopicity ($\kappa_{CCN}$) are between ~0.1 and ~0.2 and show minor differences among the clusters. The median value of $\kappa_{CCN}$ is ~0.15 for all four clusters,





substantially below 0.3 suggested for continental aerosols. Organic hygroscopicity $\kappa_{org}$ was derived from $\kappa_{CCN}$ and aerosol chemical composition. The variation of the cluster average $\kappa_{org}$ with O:C ratio generally follows the trend reported by an earlier laboratory study (Lambe et al. 2011) and those predicted based on organic molecular weight and volatility (Wang et al., 2019).

The organic aerosols observed during ALC-IOP were previously classified into three SOA factors using PMF analysis (Zhou et al., 2016). Hygroscopicities of the three factors ($\kappa_{LV\text{-}OOA}$, $\kappa_{SV\text{-}OOA}$, and $\kappa_{NOA}$) are retrieved using multilinear regression of $\kappa_{org}$ to the volume fractions of the organic factors(Thalman et al., 2017). The retrieved $\kappa_{LV\text{-}OOA}$, $\kappa_{SV\text{-}OOA}$, and $\kappa_{NOA}$, show a general increasing trend with an increasing O:C ratio, in agreement with those derived from the earlier field study (Thalman et al., 2017) and predicted by the Wang et al. (2019).  From August 10 to 15, the CCN activities of both ambient aerosol and those

processed by a TD were measured. The derived $\kappa_{org}$ shows an initial increase as TD temperature increases from 20 ℃ (i.e., ambient temperature) to 50 or 75 ℃, then decreases as TD temperature further increases to 100 ℃. The O:C ratio follows a similar trend with the TD temperature setting. The variations of $\kappa_{org}$ and O:C with TD temperature observed during ALC-IOP are different from previous laboratory experiments and field observations, which reported that organic O:C consistently increased with increasing TD temperature (Huffman et al., 2009;Kuwata et al., 2011)   whereas $\kappa_{org}$ decreased with TD

temperature(Cain and Pandis, 2017). The initial increases of O:C and $\kappa_{org}$ at TD temperature below 50 ℃ are likely due to the evaporation of more volatile organics with relatively lower O:C hygroscopicity, such as primary OA. Above ~50℃, evaporated organics are less oxygenated and have a lower molecular weight (i.e., molecule size) compared to those remaining in the particle phase.

**Acknowledgments**

This study was funded by the US DOE Atmospheric Radiation Measurement (ARM) and the Atmospheric System Research (ASR) Program, Grant No. DE-FG02-11ER65293, DE-SC0007178, and used data from the ARM Climate Research Facility (a DOE Office of Science User Facility). Shan Zhou was partially funded by the Donald G. Crosby Fellowship and the Fumio Matsumura Memorial Fellowship at UC Davis. We acknowledge the US Department of Energy Atmospheric Radiation

Measurement program and Brookhaven National Laboratory for logistics support.

**Data availability**
The CCN data in the study are available upon reasonable request to Jian Wang (jian@wustl.edu). The other data are available from the ARM data archive: https://www.arm.gov/research/campaigns/osc2011aerosolife

**Author contributions**
JW and FM designed the research. FM, QZ, JX, and JW carried out the measurements. FM, SZ, SC, JW led the analyses, and FM led the writing, with major input from JW and further input from all other authors.

**Competing interests**
The authors declare that they have no conflict of interest.



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
