# Peer review of "Measurement report: CCN activity and its variation with organic oxidation level and volatility observed during aerosol life cycle intensive operational period (ALC-IOP)"

_Atmospheric Chemistry and Physics, 2021_

## Author Comment (AC1)

This manuscript by Mei et al. studied the CCN activity and its variation with organic oxidation level (O:C ratio) and volatility at a rural site on Long Island, New York. It showed that aerosols originated from the Pennsylvania and New York areas contained more CCN active particles than other air mass. The hygroscopicity of organics (kappa_org) and O:C ratio first increase and then decrease with the thermal denuder (TD) temperature because of the evaporation of gas species with different chemical compositions. Overall, the manuscript is well written, and it offers new perspectives on CCN activity and the influencing parameters. I recommend the publication of the manuscript after minor revisions.

**Response: We sincerely appreciate the comments and suggestions from our reviewer. Thank you very much for considering the publication of our manuscript. We address your comments below (also in blue).**

1. Line 28: It seems that the last sentence of the abstract is disconnected from the context. Also, it is necessary to briefly discuss the reasons for the decrease of O:C and kappa_org here.

   **Response: Thank you very much for the suggestions. We modified the abstract line 24-31. "The derived $\kappa_{Org}$ and O:C ratio first increase as TD temperature increases from 20 ℃ (i.e., ambient temperature) to 50 or 75 ℃, then decrease as TD temperature further increases to 100 ℃. The initial increases of O:C and $\kappa_{Org}$ with TD temperature below 50 ℃ are likely due to evaporation of more volatile organics with relatively lower O:C and hygroscopicity such as primary OA. At the high TD temperatures, the decreases of O:C and $\kappa_{Org}$ indicate that evaporated organics were more oxygenated and had lower molecular weight. These trends are different from previous laboratory experiments and field observations, which reported that organic O:C increased monotonically with increasing TD temperature whereas $\kappa_{Org}$ decreased with the TD temperature. One possible reason is that previous studies were either focused on laboratory generated SOA or based on field observations at locations dominated by SOA. "**

2. Line 110: The information for the SMPS is also discussed in Line 130. This sentence can be removed. Note that the format for the reference in Line 130 needs to be corrected.

   **Response: Line 151 (previous 110) talks about the size distribution measured by the size-resolved CCN system, which was used to drive the size-resolved activation fraction. Line 171 (previous 130) talks about standalone size distribution measured by SMPS. We have modified the manuscript to clarify.**

3. Fig. 1: Please note that the format of the y-axes in this figure needs to be corrected. Also, the fonts for the axis labels are too small. Fig. 5 has the same issues.

   **Response: There is a conversion issue in the pdf file, we have set the fonts to 14 for Fig. 1 and 5. We will work with editor to make sure the font sizes are correct in the final published version.**

4. Line 164: Please check whether it is 0.50% or 0.48% as indicated in the methods section. There are a few other places in the manuscript using 0.50%.

   **Response: Thank you very much for catching the error here. It is 0.50% and we have changed the line 202.**

5. Line 171: Should this be 0.48% instead of 0.12%? Also, for the following sentence, why would the result show a strong size dependency? Would chemical composition play a role as well?

   **Response: Thank you for the correction. We corrected the line 178. We also revised the sentence in line 178-181 to "Figure 1b indicates that the airmasses in W and SSW clusters also had the highest average accumulation mode number concentration, consistent with the influence of urban emissions in Pennsylvania and New York metropolitan areas.  This trend also agrees with that of aerosol mass loading measured by the AMS (Zhou et al., 2016). In contrast, the 25%- 75% percentiles of CCN concentrations at 0.5% supersaturation ranged from 1,000 to 2,500 cm$^{-3}$ for all clusters, indicating no clear trend with air mass as observed for CCN concentration at 0.12%. These results suggest different CCN spectrum profiles (i.e., dependence of CCN concentration on SS) for aerosols from different regions."**

6. Line 174: Should "only for the larger accumulation mode aerosol particles" be removed?

   **Response: Revised as shown in the response in the comment 5.**

7. Line 209: How many periods are there for each of the four air mass clusters?

**Response: We have 10 periods influenced by the air mass cluster from NW, and 17 periods influenced by the SSW air mass cluster. The LRNW air mass cluster contributed to 8 periods and the W air mass cluster dominated 3 periods. The rest of the periods were influenced by the North Atlantic Ocean cluster and didn't included in the further discussion. We revised the manuscript and included the further information (line 294-297).**

8. Fig. 2: It shows in Fig. 3 that LV-OOA and SV-OOA are the major factors in the organic aerosols, but why are the ALC-IOP data points outside the range of LV-OOA and SV-OOA in terms of O:C atomic ratio? It may be more informative to show all the 35 data points for ALC-IOP in this figure. The label "SV-OOA" may need to be moved to a lower location in this figure.

**Response: We updated the Fig. 2 and also included all the ALC-IOP data points below. We have 25 periods which have some contributions (<20%) from NOA. In addition, the fractions of different factors (e.g., LV-OOA and SV-OOA) shown in Fig. 3 were derived from bulk aerosol composition, whereas the O:C values of ALC-IOP data points shown in Fig. 2 are averaged for particles with diameter ranging from 103 to 181 nm (e.g., the size range of size-resolved CCN measurements). Due to the above two reasons, the averaged O:C based on the air mass clusters have slightly lower values (Fig. 2).**

Fig. 4: What are the air mass sources for the events in this figure? In the figure caption, please change "black circle" to "black dot" for consistency.

**Response: We changed the caption in line 350 - 351 and also added the air mass sources.**

9. Line 333: Above ~ 50 Celsius, the evaporated organics should be more oxygenated (not less oxygenated), which would lead to a lower O:C ratio.

**Response: Sorry for the typo. Yes, the reviewer is correct. Above 50 Celsius, the evaporated organics were on average more oxygenated than those remaining in the particle phase. We had revised the sentence in line 410. "Above ~50℃, evaporated organics are on average more oxygenated and have a lower molecular weight (i.e., molecule size) compared to those remaining in the particle phase."**

---

## Author Comment (AC2)

The manuscript by Mai et al. presents new measurement data and provide detailed and relevant information about aerosol hygroscopicity measured at Long Island. It is well written and have a high quality. It should be published after the following comments are considered.

**Response: We sincerely appreciate the comments and suggestions from our reviewer. Thank you very much for considering the publication of our manuscript. We address your comments below (also in blue).**

1) I would recommend to make all data publicly available without request is needed. Because the purpose of a measurement report is that other people can work with the data in future.

**Response: We have all the raw data archived by the ARM Data Center and publicly available for the community. Thus, we revised the data availability session.**

2) I would love to see a map with the trajectory analysis maybe in the main text or in the supplement.

**Response: The trajectory analysis map was shown in Zhou's paper. Thus, we added one sentence to clarify (line 153-154).**